# All of gene expression (AOE): An integrated index for public gene expression databases

**Hidemasa Bono** [ID] *

Database Center for Life Science (DBCLS), Joint Support-Center for Data Science Research, Research Organization of Information and Systems, Mishima, Japan

* bono@dbcls.rois.ac.jp

## Abstract

Gene expression data have been archived as microarray and RNA-seq datasets in two public databases, Gene Expression Omnibus (GEO) and ArrayExpress (AE). In 2018, the DNA DataBank of Japan started a similar repository called the Genomic Expression Archive (GEA). These databases are useful resources for the functional interpretation of genes, but have been separately maintained and may lack RNA-seq data, while the original sequence data are available in the Sequence Read Archive (SRA). We constructed an index for those gene expression data repositories, called All Of gene Expression (AOE), to integrate publicly available gene expression data. The web interface of AOE can graphically query data in addition to the application programming interface. By collecting gene expression data from RNA-seq in the SRA, AOE also includes data not included in GEO and AE. AOE is accessible as a search tool from the GEA website and is freely available at https://aoe.dbcls.jp/.

**Data Availability Statement:** Availability of supporting source code, Project name: All Of gene Expression (AOE) Project home page: https://aoe.dbcls.jp/en GitHub page: https://github.com/dbcls/AOE/ Operating system: UNIX (macOS and linux) Programming language: Python, Perl License: MIT.

## Introduction

After the invention of microarray, it became possible to measure the abundance of all transcripts at the genomic scale, which is now called the transcriptome. Since then, gene expression data from those experiments have been archived in public repositories after the development of the Minimum Information About a Microarray Experiment (MIAME) standard [1]. These are the NCBI Gene Expression Omnibus (GEO; https://www.ncbi.nlm.nih.gov/geo/) [2] and the EBI ArrayExpress (AE; https://www.ebi.ac.uk/arrayexpress/) [3] in a MIAME compliant manner.

Unlike the International Nucleotide Sequence Database [4], these two databases for gene expression have not been exchanging data with each other. AE had once imported data from GEO but stopped doing so in 2017 (https://www.ebi.ac.uk/arrayexpress/help/GEO_data.html). Archived GEO data is still available from AE, but new data archived in GEO is no longer available from AE. Therefore, users need to search both databases to get comprehensive public gene expression data of interest because these databases have been separately maintained. Furthermore, the DNA DataBank of Japan (DDBJ) recently started a similar repository called the Genomic Expression Archive (GEA; https://www.ddbj.nig.ac.jp/gea/) [5]. Hence there is a need for integration of these public gene expression databases.

Availability of supporting data: The constructed data are archived at the Life Science Database Archive at the National Bioscience Database Center (NBDC), Japan Science and Technology Agency (JST), and is available with the DOI: 10.18908/lsdba.nbdc00467-000 (https://doi.org/10.18908/lsdba.nbdc00467-000). License: CC-BY 4.0.

**Funding:** This work was supported by the National Bioscience Database Center (NBDC), Japan Science and Technology Agency (JST) to HB. The funders had no role in study design, data collection and analysis, decision to publish, or preparation of the manuscript.

**Competing interests:** The authors have declared that no competing interests exist.

Also, these databases may lack transcriptome sequencing data (RNA-seq) while the original sequence data are accessible in the nucleotide sequence repository of high-throughput sequencing platforms; the Sequence Read Archive (SRA) [6]. This is because data deposition to GEO and AE is not mandatory when the original sequencing data are deposited to the SRA.

We, therefore, developed an index of public gene expression databases, called All Of gene Expression (AOE). The aim of AOE is to integrate gene expression data and make them all searchable together. We have maintained AOE for five years, and it has been useful for functional genomics research. Here, we report a detailed description and utility of AOE. AOE is freely accessible from https://aoe.dbcls.jp/.

## Results

### Status of gene expression databases

Gene expression data in NCBI Gene Expression Omnibus (GEO) used to be continuously imported into EBI ArrayExpress (AE), and thus we were theoretically able to obtain all data deposited to GEO from AE. Therefore, All Of gene Expression (AOE) was originally indexed for AE only.

Unfortunately, AE discontinued GEO data import in 2017. At that point, we investigated data-series entries in these two databases by matching GEO series IDs: IDs beginning with GSE in GEO and those beginning with E-GEOD in AE; for example, GSE52334 in GEO corresponds to E-GEOD-52334 in AE. Apparently over thirty thousand entries were missing in AE (Fig 1). Furthermore, even GEO did not publicly represent the whole transcriptome data, as over ten thousand entries in AE were missing in GEO. Thus, we decided to include those missing entries in AOE. In other words, we started indexing GEO data and other public transcriptome data, including the DDBJ Genomic Expression Archive (GEA), to allow all public gene expression data to be searched.

### An index of gene expression data series from metadata

AOE was originally developed to provide a graphical web interface to search EBI AE, which is one of the public gene expression databases described above. We call this dataset that includes data from AE only 'AOE level 1' (Fig 2). Data at this level contain only IDs for AE, and the entries imported from GEO contain IDs for both BioProject and GEO.

After the import of GEO data to AE was discontinued, AOE began importing GEO data by directly utilizing the DBCLS SRA application programming interface (API) [7]. By subtracting the GEO data already existing in AE, new entries were included in AOE. We call the merged dataset that includes GEO data 'AOE level 2' (Fig 2). Data at this level contain IDs for BioProject and GEO, but not for AE.

There were still some gene expression data missing that were not included in AE and GEO but were registered as transcriptome sequencing data in SRA. The final merged dataset is called 'AOE level 3' and represents a real public gene expression dataset (Fig 2). Data at this level contain BioProject IDs only.

Fig 2 shows a schematic view of the data flow that was used to prepare the index data for AOE, as summarized above. The constructed data are archived in the Life Science Database Archive at the National Bioscience Database Center (NBDC), Japan Science and Technology Agency (JST), and is available at the DOI: 10.18908/lsdba.nbdc00467-000 (https://doi.org/10.18908/lsdba.nbdc00467-000).

As AOE was designed to index public gene expression data, 'experimental series'-wise data have been indexed for the search. Individual hybridization data for microarray and run data for RNA-seq are directly linked to the original databases. All codes to parse public databases

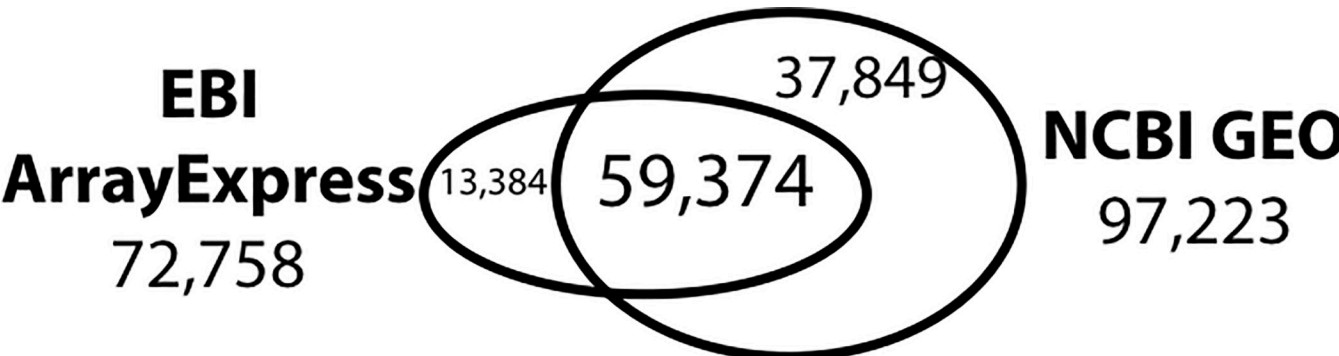

**Fig 1. Comparison of EBI ArrayExpress and NCBI Gene Expression Omnibus.** The number of overlapping data-series entries in ArrayExpress (left) and Gene Expression Omnibus (right).

and construct a web service are accessible from the DBCLS AOE GitHub repository (https://github.com/dbcls/AOE/). These are free and open-source software and can be installed anywhere.

### Graphical web interface

Gathering all three levels of data described above, AOE enables visualization and exploration of gene expression data. AOE provides an interactive web interface (https://aoe.dbcls.jp/) to retrieve data of interest (Fig 3). Users can see overall statistics of stored data in AOE (Fig 3A). The histogram for ranking by quantification methods can be dynamically created by clicking on the technology name. Fig 3B shows the number of data in AOE only for sequencing assays (RNA-seq).

Users can easily filter data by organism and quantification method of gene expression. For example, users can search with the keyword 'hypoxia' (Fig 3A). AOE currently reports 524 items with three histograms (by year, organism, and quantification method; Fig 3C). After looking at the histograms, the user can filter the data by '*Homo sapiens*' by dragging the bar in the histogram by organism. Then, AOE recreates the histograms with the selected data (Fig 3D). Additionally, the user can filter the data by 'Illumina' by dragging the bar in the histogram by quantification method (Fig 3E). The selected data (58 records currently) can be retrieved by clicking on the 'Retrieve' button (Fig 3F). Users can browse retrieved data and jump to original data by clicking on IDs in the table (ArrayExpress, BioProject, and GEO; Fig 3G). Optionally, users can also download the list of IDs from the 'Download ID list' button.

A shortcut to retrieve a list of specific organisms is to click on the species icon with nomenclature and the 'retrieve' button (Fig 3A). The top 30 species in AOE will be listed and can be accessed in this way.

Further, a tutorial movie that shows how to make use of the AOE web interface is available at https://doi.org/10.7875/togotv.2018.146. The movie originated from the contents of TogoTV, which provides tutorial videos for useful databases and web tools in the life sciences [8], and is available on the TogoTV original website (https://togotv.dbcls.jp/en/) and YouTube (https://youtube.com/togotv/).

AOE web server has been maintained for five years. From the usage statistics (from July 2015 to Oct 2019), there were 95,334 visits, 393,174 page views, and 630,837 hits. The fact that two-thirds of visits were under 30seconds indicates that users accessed the AOE web server in their web browser with an instant query with specific keywords.

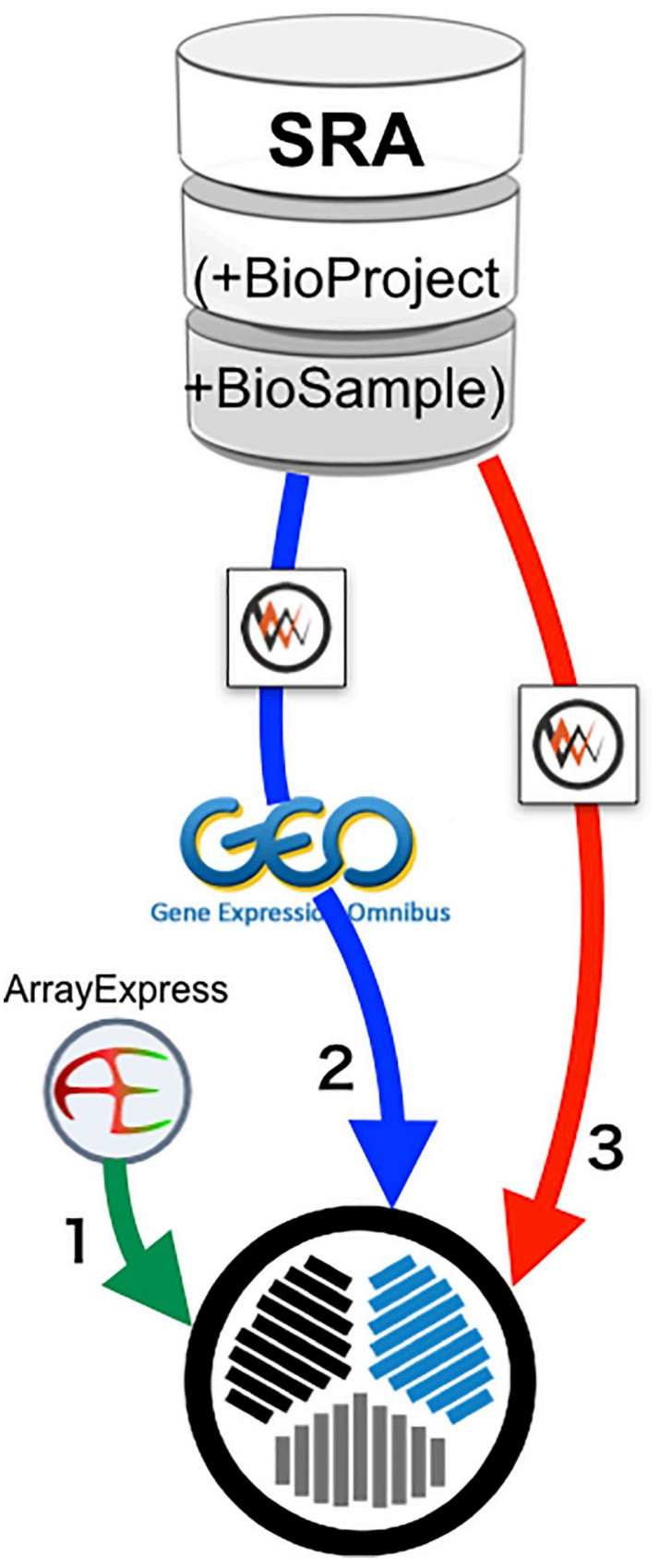

**Fig 2. How to make AOE index.** Three levels of AOE data flow are required to make the AOE index. Level 1 from ArrayExpress data, level 2 from GEO data in SRA via DBCLS SRA API, and level 3 from RNA-seq data in SRA, but not in GEO via DBCLS SRA API.

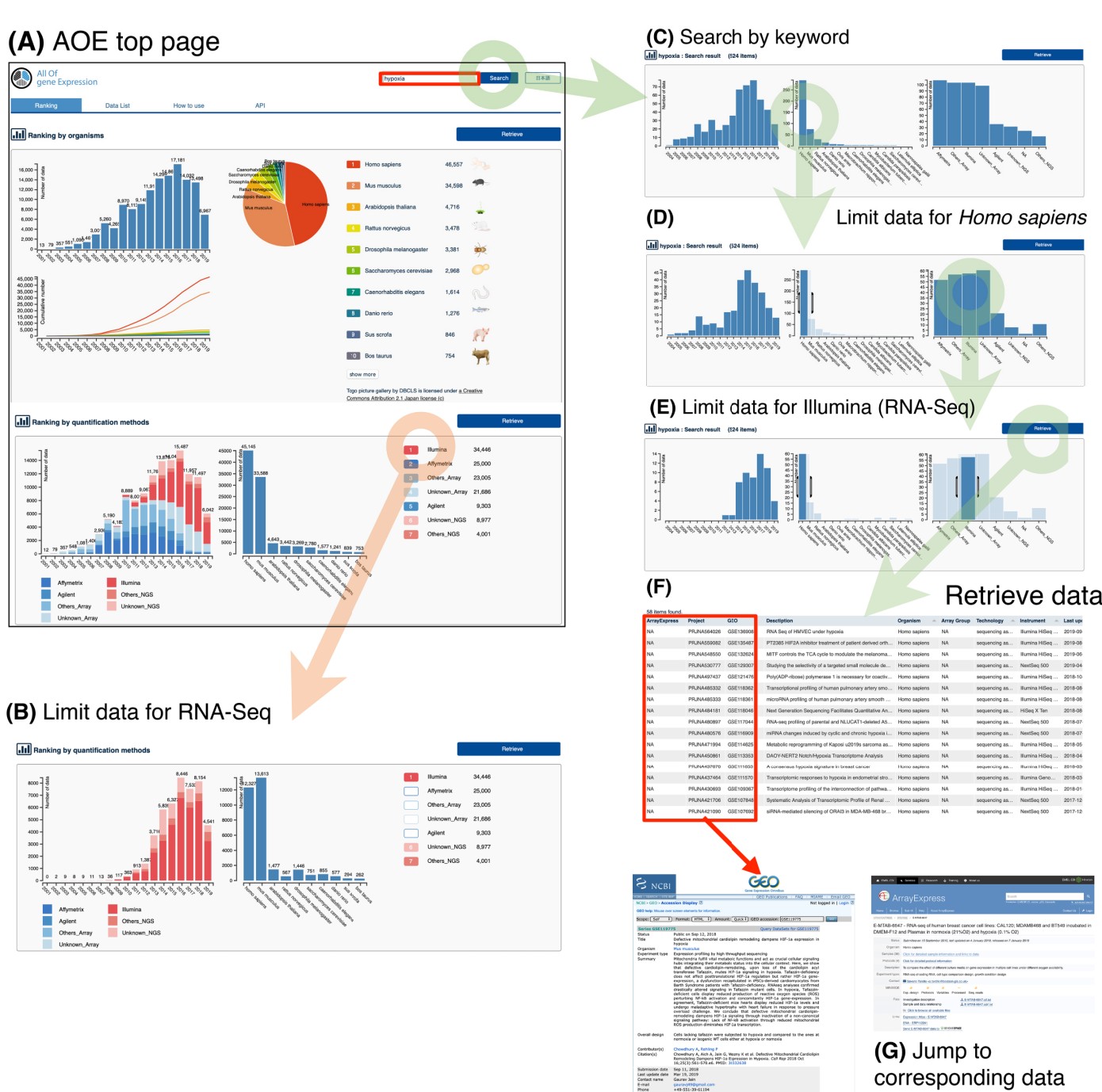

**Fig 3. AOE web interface.** Users can retrieve data of interest graphically in the AOE web interface.

## Application programming interface

Users can also query AOE via API. AOE provides a simple Representational State Transfer (REST) API that enables users to perform searches with their client programs in an automated manner. The search results in a JSON formatted output can be retrieved through the following URL:

https://aoe.dbcls.jp/api/search?fulltext=KEYWORD&[Technology= TECHNOLOGY&Organisms=ORGANISM&page=OFFSET&size=SIZE]

KEYWORD: keyword to search AOE

TECHNOLOGY: technology to use expression profiling (sequencing, microarray, Affymetrix, Agilent, Illumina, etc.)

ORGANISM: 'homo%20sapiens' for human, 'mus%20musculus' for mouse, etc.

OFFSET: the page number

SIZE: the number of results in one page

1. A search for the keyword 'hypoxia' with 25 results in one page is represented as follows: https://aoe.dbcls.jp/api/search?fulltext=hypoxia&page=1&size=25

2. A search for human data in RNA-seq (sequencing assay) with twenty-five results in one page is represented as follows:

https://aoe.dbcls.jp/api/search?Technology=sequencing&organisms=homo% 20sapiens&page=1&size=25

A precise description for the AOE API is available from the AOE website or directly from the DBCLS AOE GitHub website at https://github.com/dbcls/AOE/blob/master/API_ documentation.md.

## Discussion

We have developed and maintained an index of public gene expression databases, called All Of gene Expression (AOE). AOE originally began as an index for the ArrayExpress (AE) database maintained at EBI (we call this 'AOE level 1'), because AE had exported gene expression data from Gene Expression Omnibus (GEO), which is the largest gene expression database maintained at NCBI. That meant AE contained all gene expression data, including those deposited to GEO.

AE stopped importing data from GEO in 2017. While GEO data archived in AE is still available from AE, new data archived in GEO is no longer available from AE. Thus, we started indexing GEO data directly by making use of the API of DBCLS SRA (AOE level 2). In 2018, the DNA DataBank of Japan (DDBJ) started the Genomic Expression Archive (GEA), which is a repository for gene expression quantification data. Integration of these public gene expression databases is needed to increase the reusability of gene expression data. Newly submitted data contain BioProject IDs, and this feature makes it possible to integrate multiple levels of indices and resolve complicated relationships among IDs, while old AE entries do not have BioProject ID.

The existence of a great deal of data at AOE level 3 shows that not all sequencing gene expression data are stored in GEO. This indicates that GEO is insufficient as a complete public gene expression database. Much of the data at AOE level 3 are heterogeneous, and metadata for those can lack several descriptions, which are curated and cleanly described in GEO and AE.

A similar approach has also been undertaken by EBI, called the Omics Discovery Index (OmicsDI; https://www.omicsdi.org/), which provides a knowledge discovery framework across heterogeneous omics data (genomics, proteomics, transcriptomics, and metabolomics)

[9]. OmicsDI aims to integrate various types of omics data and is not focused on gene expression data.

AOE is focused on gene expression data. It is also designed to be a search interface for DDBJ GEA, and a link to AOE can be found at the official GEA website. When AOE is used as a search interface for DDBJ GEA, it is expected that AOE will be continuously used at the DDBJ website.

The web interface for AOE is simple and user-friendly, and so AOE can also be used by biologists who are not familiar with database searching. AOE can also be used by professionals to construct reference expression datasets for specific organisms. We have also developed a Reference Expression dataset (RefEx) for humans and mice [10]. We are planning to implement RefEx for other organisms, by making use of these reference expression datasets retrieved by AOE.

For future development, we are planning to use not only metadata but also quantified expression data that will allow users to search for data based on the similarity of gene expression profiles. Moreover we are going to use the quality control results from the FASTQ program to screen for RNA-seq data.

## Methods

### Acquisition of public gene expression data

AOE consists of two major types of data sources. One is the EBI ArrayExpress (AE), and the other is data in NCBI, including the Gene Expression Omnibus (GEO).

For the AE data type, several files are required to make an AOE index. These files are in a simple spreadsheet-based, MIAME-supportive format, called MicroArray Gene Expression Tabular (MAGE-TAB) files, which are Array Design Format (ADF), Investigation Description Format (IDF), and Sample and Data Relationship Format (SDRF) files [11]. These files are routinely acquired from the AE FTP site (ftp://ftp.ebi.ac.uk/pub/databases/arrayexpress/data/). ADF files are located in subdirectories in ftp://ftp.ebi.ac.uk/pub/databases/arrayexpress/data/array/, with the file extension .adf.txt. IDF and SDRF files are located in subdirectories in ftp://ftp.ebi.ac.uk/pub/databases/arrayexpress/data/experiment/, with file extensions .idf.txt and .sdrf.txt, respectively.

For data from NCBI, in addition to the file describing ID relationships in the Sequence Read Archive (SRA) named SRA_Accessions.tab from ftp://ftp.ncbi.nlm.nih.gov/sra/reports/Metadata/, metadata from SRA, BioProject and BioSample are used to make an index for AOE, as all data from GEO have BioProject IDs and BioSample IDs even if the gene expression quantification for that data was not a transcriptome sequencing one.

### Organizing metadata from different sources

For the AE type of data, ADF, IDF, and SDRF files are required to make an index for AOE. Data from the DDBJ Genomic Expression Archive (GEA) also consist of the AE type of data and are available from its FTP site (ftp://ftp.ddbj.nig.ac.jp/ddbj_database/gea/). We used the AE type of data to construct an initial AOE index set (called AOE level 1).

GEO data in the Sequence Read Archive (SRA), BioProject, and BioSample are used to make an index for AOE. These data have been stored in the DBCLS SRA as JSON-LD, and the application programming interface (API) for metadata for those has also been maintained in DBCLS. AOE used this API to retrieve data needed to make the index (AOE level 2).

Finally, we collected the RNA-seq data in SRA, making use of the DBCLS SRA API. Most of this fraction of data are in AOE level 2, but many entries can be found in this filter (AOE level 3).

A concatenated tab-delimited file of the constructed index is archived in the Life Science Database Archive at National Bioscience Database Center (NBDC), Japan Science and Technology Agency (JST) at DOI: 10.18908/lsdba.nbdc00467-000 (https://doi.org/10.18908/lsdba.nbdc00467-000).

Data parsers to make a tab-delimited text file for visualization are implemented in Perl5 and UNIX shell commands. All shell and Perl5 scripts for those are accessible from GitHub (https://github.com/dbcls/AOE/).

## Visualization of datasets

For visualizing datasets, we implemented specially coded Python3 scripts, and we also used D3.js, a JavaScript library for manipulating documents based on data (https://d3js.org/). This enables data selection by mouse operation. For example, the user can select data by release date by dragging the histogram generated with the keyword search.

## Acknowledgments

The author wishes to thank Dr. Naoya Oishi for the design and development of the web and application programming interface for AOE, Dr. Yuichi Kodama for helpful advice for the integration of the DDBJ Genomic Expression Archive (GEA), and Dr. Takeru Nakazato and Dr. Tazro Ohta for their help in using the DBCLS SRA API for updating AOE entries from NCBI databases. The tutorial movie for the AOE web interface was created under the TogoTV project in DBCLS with editorial direction by Dr. Hiromasa Ono. The computing resource was partly provided by the supercomputer system at the National Institute of Genetics (NIG), Research Organization of Information and Systems (ROIS), Japan. We would like to thank Editage (www.editage.com) for English language editing.

## Author Contributions

**Conceptualization:** Hidemasa Bono.

**Data curation:** Hidemasa Bono.

**Funding acquisition:** Hidemasa Bono.

**Investigation:** Hidemasa Bono.

**Project administration:** Hidemasa Bono.

**Supervision:** Hidemasa Bono.

**Validation:** Hidemasa Bono.

**Visualization:** Hidemasa Bono.

**Writing – original draft:** Hidemasa Bono.

**Writing – review & editing:** Hidemasa Bono.

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
