## [Decision Letter · Decision Letter 0]

7 Oct 2019

PONE-D-19-23442

All of gene expression (AOE): integrated index for public gene expression databases

PLOS ONE

Dear Dr. Bono,

Thank you for submitting your manuscript to PLOS ONE. After careful consideration, we feel that it has merit but does not fully meet PLOS ONE’s publication criteria as it currently stands. Therefore, we invite you to submit a revised version of the manuscript that addresses the points raised during the review process.

We would appreciate receiving your revised manuscript by Nov 21 2019 11:59PM. To enhance the reproducibility of your results, we recommend that if applicable you deposit your laboratory protocols in protocols.io, where a protocol can be assigned its own identifier (DOI) such that it can be cited independently in the future. For instructions see: http://journals.plos.org/plosone/s/submission-guidelines#loc-laboratory-protocols

We look forward to receiving your revised manuscript.

Kind regards,

Robert Hoehndorf, Ph.D.

Academic Editor

PLOS ONE

Journal Requirements:

1. Please amend either the title on the online submission form (via Edit Submission) or the title in the manuscript so that they are identical.

Additional Editor Comments (if provided):

The reviewers have considered the manuscript. They have raised a number of points that should be taken into consideration in a revision. Many changes are minor, but Reviewer 2 asks for substantially more information to be included on the implementation and the use of the services developed, which will improve the utility of the resource and increase adoption; this should be addressed in the revised manuscript.

Reviewers' comments:

Reviewer's Responses to Questions

**Comments to the Author**

1. Is the manuscript technically sound, and do the data support the conclusions?

Reviewer #1: Yes

Reviewer #2: Yes

2. Has the statistical analysis been performed appropriately and rigorously? 

Reviewer #1: N/A

Reviewer #2: N/A

3. Have the authors made all data underlying the findings in their manuscript fully available?

Reviewer #1: Yes

Reviewer #2: Yes

4. Is the manuscript presented in an intelligible fashion and written in standard English?

Reviewer #1: Yes

Reviewer #2: Yes

5. Review Comments to the Author

Reviewer #1: These are more of recommendations to the authors regarding the manuscript:

1. For the statement: "Unlike the International Nucleotide Sequence Database [3], these two databases for gene expression have stopped exchanging data with each other since AE stopped importing data from GEO"

Is there a reason why they stopped exchanging data? Could this be elaborated further, for the benefit of the readers?

2. This paper highlights the creation of an index, but could it be possible to describe how the index was generated and what technology was being used?

3. The authors note that AOE has been around for 5 years. Could it be possible for the authors to share usage statistics over this period that can highlight usage patterns. This might also be insightful to readers.

Reviewer #2: * All of Gene Expression (AOE) by Bono

** Editorial overview

The author created tools that mine metadata from the main public gene

expression databases and created a queryable cross-database index 'all

of gene expression' (OAE) that can be accessed through an API and a

browser web interface. This paper describes the need for such an index

and presents it as a completed work.

I think the work is worthwhile publishing but the publication needs

some additional work to make it more interesting for the reader. The

paper is very short as it is, and I think it can be improved by

providing more context and useful examples.

** Notes

Some historical context is missing. MIAME is not mentioned though

(probably) used. Similar efforts at describing expression

experiments should be mentioned and put in context. Likewise

XML metadata descriptions exist and RDF ontologies. I miss that

information and some sense of how important they are to providing

and exposing an index of gene expression experiments. Does the

current work provide metadata, what does it look like and why is

it (not) RDF?

In the discussion the author can mention the challenges of matching

data and data quality. In microarrays the data files come in different

forms - especially raw data, differential expression and normalized

data. How do we deal with that when mining data? As it stands the

expression databases are highly suspect. An index should give hints

about the state of the data.

A different topic concerns probe-level information and RNA-seq

alignments using different tools and different reference

genomes. These are fraught with problems and naive comparisons are

pretty useless. Providing an index to data resources raises questions

about what these resources contain and how to deal with that. I

realise an index is a starting point, but maybe the author can explain

what the next steps are?

I think the paper would greatly benefit from a description of the REST

API and by giving some examples using R or Python. This may lead

to readers trying out the API. It also will explain the value of

the index for software developers.

Similarly, a few figures of the browser GUI that *explain* the use of

it would be helpful. The provided figure is not descriptive and even

lacks a caption.

Finally I think there should be a clear statement that all source code

is free and open source software and can be installed using this and

that...

Even more important: as a web service is provided, there should be

indication about future maintenance plans, and that the data and

service can be replicated freely elsewhere (and how it can be

done). There are too many bioinformatics initiatives that disappear

after publication. How does the author want to warrant continuous

service?

The paper states that the 'parser' source code is available. What

about the web-service software? It is not clear from the github repo

either. Interestingly the source code points out the use of the

common workflow language (CWL). That, I believe, should be highlighted

too.

** English editing

- I prefer not to start sentences with 'However'

- Findings in abstract I would use Results instead - because

these are useful tools

- Conclusions or Conclusion?

** Conclusion

The paper would greatly benefit from extra information. No additional

software implementation by the author is required, my comments only

relate to presentation and making the work more interesting/useful for

the reader. I would say it is a minor revision, though as the paper is

expected to double in size it may be considered major.

6. PLOS authors have the option to publish the peer review history of their article (what does this mean?). If published, this will include your full peer review and any attached files.

Reviewer #1: No

Reviewer #2: Yes: Pjotr Prins

---

## [Author Response · Author response to Decision Letter 0]

11 Nov 2019

Detailed point-by-point response two sets of referees’ comments are in the rebuttal letter in the separate file, but I will paste the contents of our responses below.

Response to Reviewer #1

> 1. For the statement: "Unlike the International Nucleotide Sequence Database [3], these two databases for gene expression have stopped exchanging data with each other since AE stopped importing data from GEO"

> Is there a reason why they stopped exchanging data? Could this be elaborated further, for the benefit of the readers?

Thanks for your suggestion. 

In the web page of ArrayExpress about GEO data (https://www.ebi.ac.uk/arrayexpress/help/GEO_data.html), they just say

> “We have stopped the regular imports of Gene Expression Omnibus (GEO) data into ArrayExpress. We will keep using data from GEO to build our added value database Expression Atlas, and the reprocessed and additionally annotated data for selected datasets will be available from there.”

and no reason for that is described. We do not know the reason for that. 

Thus, in the revised manuscript, we added the URL above and one more sentence emphasizing new data in GEO are not included in AE.

“AE once had imported data from GEO, but stopped importing data in 2017 (https://www.ebi.ac.uk/arrayexpress/help/GEO_data.html). Archived GEO data is still available from AE, but new data archived in GEO no longer available from AE.” in the Introduction section and “AE stopped importing data from GEO in 2017. While GEO data archived in AE is still available from AE, new data archived in GEO no longer available from AE.” in the Discussion section.

> 2. This paper highlights the creation of an index, but could it be possible to describe how the index was generated and what technology was being used?

Thanks again for your suggestion. 

How to construct AOE index is described in “An index of gene expression data series from metadata” subsection in the Result section and depicted as Fig 2.We added Fig 2 pointers for this subsection. 

For the technology issue,as describe in Method section (Organizing metadata from different sources), we use shell commands and Perl script for data parsing.

“Data parsers to make a tab-delimited text file for visualization are implemented in Perl5 and UNIX shell commands. All shell and Perl5 scripts for those are accessib

le from GitHub (https://github.com/dbcls/AOE/).”

For the web interface, also described in the following section, we used Python3 codes and D3.js javascript library. “For visualizing datasets, we employed specially coded Python3 scripts, and we also used D3.js, a JavaScript library for manipulating documents based on data (https://d3js.org/)."

> 3. The authors note that AOE has been around for 5 years. Could it be possible for the authors to share usage statistics over this period that can highlight usage patterns. This might also be insightful to readers.

Thanks for your constructive suggestion. 

We added the description about usage statistics in the Results section.

“AOE web server has been maintained for five years. From the usage statistics (during July 2015 to Oct 2019), there were 95,334 visits, 393,174 page views and 630,837 hits. From the fact that most visits were under 30seconds, it seems that users accessed AOE web server in their web browser with an instant query with specific keywords.”

Response to Reviewer #2

> I think the work is worthwhile publishing but the publication needs

> some additional work to make it more interesting for the reader. The

> paper is very short as it is, and I think it can be improved by

> providing more context and useful examples.

Thank you very much for your constructive comments. 

We added more contents and some examples to use AOE to the revised manuscript.

> ** Notes

>

> Some historical context is missing. MIAME is not mentioned though

> (probably) used. 

Description about Minimum Information About a Microarray Experiment (MIAME) standard was added in the Introduction according to your suggestion.

“Since then, gene expression data from those experiments have been archived in public repositories after the development of the Minimum Information About a Microarray Experiment (MIAME) standard [1].”

Indeed, IDF, SDRF and ADF files are used to make AOE index, and these metadata are described as tab delimited file known as MAGE-TAB. Thus we added this point in the Methods section.

“These files are in a simple spreadsheet-based, MIAME-supportive format, called MicroArray Gene Expression Tabular (MAGE-TAB) files, which are Array Design Format (ADF), Investigation Description Format (IDF), and Sample and Data Relationship Format (SDRF) files”

> Similar efforts at describing expression experiments should be mentioned and put in context. 

Concerning a similar effort, EBI holds the service called OmicsDI.

It was described in the Discussion section.

“A similar approach has also been undertaken by EBI, called the Omics Discovery Index (OmicsDI; https://www.omicsdi.org/), which provides a knowledge discovery framework across heterogeneous omics data (genomics, proteomics, transcriptomics and metabolomics) [9]. OmicsDI aims to integrate various types of omics data and is not focused on gene expression data.”

> Likewise XML metadata descriptions exist and RDF ontologies. I miss that

> information and some sense of how important they are to providing

> and exposing an index of gene expression experiments. Does the

> current work provide metadata, what does it look like and why is

> it (not) RDF?

AOE is an index of gene expression data, and thus not providing metadata of the contents.

Concerning the relationship to RDF, AOE is an application that makes heavy use of metadata of the Sequence Read Archive in RDF (JSON-LD formatted data). We added the description about JSON-LD in the Methods section.

“These data have been stored in the DBCLS SRA as JSON-LD, and the application programming interface (API) for metadata for those has also been maintained in DBCLS."

> In the discussion the author can mention the challenges of matching

> data and data quality. In microarrays the data files come in different

> forms - especially raw data, differential expression and normalized

> data. How do we deal with that when mining data? As it stands the

> expression databases are highly suspect. An index should give hints

> about the state of the data.

Using expression data itself is a next challenge of AOE project. It currently integrates and indexes metadata in the public databases. We added this issue in the Discussion section.

“For the future development, we are also planning to use not only metadata, but also quantified expression data that will allow users to search data based on the similarity of gene expression profiles.”

> A different topic concerns probe-level information and RNA-seq

> alignments using different tools and different reference

> genomes. These are fraught with problems and naive comparisons are

> pretty useless. Providing an index to data resources raises questions

> about what these resources contain and how to deal with that. I

> realise an index is a starting point, but maybe the author can explain

> what the next steps are?

We think that the quality control of data will be needed to screen the data.

As a first step, we are going to use the result of quality control by FASTQ program to screen the data for RNA-seq data. This point is also added in the Discussion section.

“And, we are going to use the result of quality control by FASTQ program to screen the data for RNA-seq data.”

> I think the paper would greatly benefit from a description of the REST

> API and by giving some examples using R or Python. This may lead

> to readers trying out the API. It also will explain the value of

> the index for software developers.

Thank you very much for great suggestion. A description about API was added in the manuscript in ‘Application programming interface’ subsection in the Result section. 

“Users can also query AOE via API. AOE provides a simple Representational State Transfer (REST) API that enables users to perform searches with their client programs in an automated manner. The search results in a JSON formatted output can be retrieve through the following URI: 

https://aoe.dbcls.jp/api/search?fulltext=KEYWORD&[Technology=TECHNOLOGY&Organisms=ORGANISM&page=OFFSET&size=SIZE]

KEYWORD: keyword to search AOE

TECHNOLOGY: technology to use expression profiling (sequencing, microarray, Affymetrix, Agilent, Illumina, etc)

ORGANISM: ‘homo%20sapiens’ for human, ‘mus%20musculus’ for mouse, etc.

OFFSET: the page number

SIZE: the number of results in one page

1. A search for the keyword ‘hypoxia’ with twenty-five results in one page is represented as follows:

https://aoe.dbcls.jp/api/search?fulltext=hypoxia&page=1&size=25

2. A search for human data in RNA-seq (sequencing assay) with twenty-five results in one page is represented as follows: 

https://aoe.dbcls.jp/api/search?Technology=sequencing&organisms=homo%20sapiens&page=1&size=25

“

> Similarly, a few figures of the browser GUI that *explain* the use of

> it would be helpful. The provided figure is not descriptive and even

> lacks a caption.

Thanks for your comment. 

According to your suggestion, Fig 3 was completely updated to instruct how to make use of AOE web interface, and descriptions for that were added in the manuscript. The caption for Fig3 was also added.

“Fig 3. AOE web interface 

Users can retrieve data of interest .graphically in the AOE web interface.”

Additionally, a tutorial movie for this operation on the web browser is also available called TogoTV as described in the Results section.

“Further, a tutorial movie to show how to make use of AOE web interface is available at https://doi.org/10.7875/togotv.2018.146."

> Finally I think there should be a clear statement that all source code

> is free and open source software and can be installed using this and

> that...

It is very important issue. Thank you very much for your comment. We added clear statement that all source code is free and open source software in the Results section.

“All codes to parse public databases and construct a web service are accessible from the DBCLS AOE GitHub repository (https://github.com/dbcls/AOE/). They are free and open source software, and can be installed anywhere.”

> Even more important: as a web service is provided, there should be

> indication about future maintenance plans, and that the data and

> service can be replicated freely elsewhere (and how it can be

> done). There are too many bioinformatics initiatives that disappear

> after publication. How does the author want to warrant continuous

> service?

As described in ‘Discussion’ section, we plan that AOE is now used as a search interface in DDBJ Genomic Expression Archive (GEA) at https://www.ddbj.nig.ac.jp/gea/index-e.html . 

We aimed AOE will be continuously used in the DDBJ website when AOE is used as a search interface to DDBJ GEA.

“AOE is focused on gene expression data. It is also designed to be a search interface for DDBJ GEA, and a link to AOE can be found on the official GEA website. When AOE is used as a search interface to DDBJ GEA, it is expected that AOE will be continuously used in the DDBJ website. “

> The paper states that the 'parser' source code is available. What

> about the web-service software? It is not clear from the github repo

> either. Interestingly the source code points out the use of the

> common workflow language (CWL). That, I believe, should be highlighted

> too.

Thank you very much for your careful examination of github repository for AOE.

Source codes for AOE web-service are now merged into the repository (https://github.com/dbcls/AOE/tree/master/Web). 

The CWL codes in the current repository are a product of Biohackathon2018. We are now trying to do CWLization of AOE index parsers at Biohackathons in the future.

> ** English editing

>

> - I prefer not to start sentences with 'However'

> - Findings in abstract I would use Results instead - because

> these are useful tools

> - Conclusions or Conclusion?

Thank you very much for your suggestions. 

We modified the manuscript according to your suggestions.

> ** Conclusion

>

> The paper would greatly benefit from extra information. No additional

> software implementation by the author is required, my comments only

> relate to presentation and making the work more interesting/useful for

> the reader. I would say it is a minor revision, though as the paper is

> expected to double in size it may be considered major.

Thank you very much for you various suggestions and comments. 

We are sure that the manuscript is now useful for readers of PLOS ONE.

---

## [Decision Letter · Decision Letter 1]

27 Nov 2019

PONE-D-19-23442R1

All of gene expression (AOE): an integrated index for public gene expression databases

PLOS ONE

Dear Dr. Bono,

Thank you for submitting your manuscript to PLOS ONE. After careful consideration, we feel that it has merit but does not fully meet PLOS ONE’s publication criteria as it currently stands. Therefore, we invite you to submit a revised version of the manuscript that addresses the points raised during the review process.

We would appreciate receiving your revised manuscript by Jan 11 2020 11:59PM. To enhance the reproducibility of your results, we recommend that if applicable you deposit your laboratory protocols in protocols.io, where a protocol can be assigned its own identifier (DOI) such that it can be cited independently in the future. For instructions see: http://journals.plos.org/plosone/s/submission-guidelines#loc-laboratory-protocols

We look forward to receiving your revised manuscript.

Kind regards,

Robert Hoehndorf, Ph.D.

Academic Editor

PLOS ONE

Additional Editor Comments (if provided):

The reviewers have assessed the manuscript and recommend the manuscript to be published once some minor issues are addressed. The reviewers commented on the language which needs some editing. Please address the comments of the reviewers and carefully edit the language used in the manuscript.

Reviewers' comments:

Reviewer's Responses to Questions

**Comments to the Author**

1. If the authors have adequately addressed your comments raised in a previous round of review and you feel that this manuscript is now acceptable for publication, you may indicate that here to bypass the “Comments to the Author” section, enter your conflict of interest statement in the “Confidential to Editor” section, and submit your "Accept" recommendation.

Reviewer #1: All comments have been addressed

Reviewer #2: All comments have been addressed

2. Is the manuscript technically sound, and do the data support the conclusions?

Reviewer #1: Yes

Reviewer #2: Yes

3. Has the statistical analysis been performed appropriately and rigorously? 

Reviewer #1: N/A

Reviewer #2: N/A

4. Have the authors made all data underlying the findings in their manuscript fully available?

Reviewer #1: Yes

Reviewer #2: Yes

5. Is the manuscript presented in an intelligible fashion and written in standard English?

Reviewer #1: Yes

Reviewer #2: No

6. Review Comments to the Author

Reviewer #1: 1. Regarding the usage statistics:

"From the fact that most visits were under 30seconds, it seems that users accessed AOE web server in their web browser with an instant query with specific keywords."

- It could also be first time visitors who do not spend much time on the page.

- Are there additional metrics available to identify such users?

2. Phrasing of sentences through out the paper:

"AE once had imported data from GEO, but stopped importing data in 2017"

- could be rephrased to "AE had once imported data from GEO, but stopped doing so in 2017."

"Integration of these public gene expression databases is required"

- could be rephrased to "There is a need for integration of these public gene expression databases."

"AOE was originally begun as an index for the ArrayExpress (AE) database maintained at EBI"

- could be rephrased to "AOE originally began as an index..."

Reviewer #2: Dear author, thank you for addressing my comments. I think the paper will benefit from some minor editing. Nothing serious, but I am sure a native speaker can be helpful to get it up to PLoS standards. Maybe the editor can recommend someone.

7. PLOS authors have the option to publish the peer review history of their article (what does this mean?). If published, this will include your full peer review and any attached files.

Reviewer #1: No

Reviewer #2: Yes: Pjotr Prins

---

## [Author Response · Author response to Decision Letter 1]

6 Dec 2019

Response to Reviewer #1

> 1. Regarding the usage statistics:

> "From the fact that most visits were under 30seconds, it seems that users accessed AOE web server in their web browser with an instant query with specific keywords."

> - It could also be first time visitors who do not spend much time on the page.

> - Are there additional metrics available to identify such users?

Thanks for clarification. We are using awstats (https://awstats.sourceforge.io) to analyze httpd log, and there is a statistics for ‘visits duration’. 

While the analyzed page can be browsed by months, percentages of ‘visit duration’ under 30s are around 66%. Below is an example for the latest statistics for that(in Oct 2019; the image is in the attached PDF, response to reviewers)

In other words, two thirds of web access are below 30s. 

Thus, we changed the description about usage statistics in the Results section.

“The fact that two-thirds of visits were under 30seconds indicates that users accessed the AOE web server in their web browser with an instant query with specific keywords.”

> 2. Phrasing of sentences through out the paper:

> "AE once had imported data from GEO, but stopped importing data in 2017"

> - could be rephrased to "AE had once imported data from GEO, but stopped doing so in 2017.”

> "Integration of these public gene expression databases is required"

> - could be rephrased to "There is a need for integration of these public gene expression databases."

> "AOE was originally begun as an index for the ArrayExpress (AE) database maintained at EBI"

> - could be rephrased to "AOE originally began as an index…"

Thank your very much for your editing. 

All issues were rephrased.

Response to Reviewer #2

> Reviewer #2: Dear author, thank you for addressing my comments. I think the paper will benefit from some minor editing. Nothing serious, but I am sure a native speaker can be helpful to get it up to PLoS standards. Maybe the editor can recommend someone.

Thank you very much for your comment.

The manuscript was re-edited by another editor, and English was very much improved.

I attached a PDF file of ‘CERTIFICATE OF ENGLISH EDITING’.

---

## [Editor Report · Decision Letter 2]

12 Dec 2019

All of gene expression (AOE): an integrated index for public gene expression databases

PONE-D-19-23442R2

Dear Dr. Bono,

We are pleased to inform you that your manuscript has been judged scientifically suitable for publication and will be formally accepted for publication once it complies with all outstanding technical requirements.

With kind regards,

Robert Hoehndorf, Ph.D.

Academic Editor

PLOS ONE
---

## [Editor Report · Acceptance letter]

10 Jan 2020

PONE-D-19-23442R2 

All of gene expression (AOE): an integrated index for public gene expression databases 

Dear Dr. Bono:

I am pleased to inform you that your manuscript has been deemed suitable for publication in PLOS ONE. Congratulations! Your manuscript is now with our production department. 

With kind regards,

on behalf of

Dr. Robert Hoehndorf 

Academic Editor

PLOS ONE